# γBMGC: A Comprehensive and Accurate Database for Screening TMAO-Associated Cardiovascular Diseases

**DOI:** 10.3390/microorganisms13020225

**Published:** 2025-01-21

**Authors:** Guang Yang, Tiantian Tao, Guohao Yu, Hongqian Zhang, Yiwen Wu, Siqi Sun, Kexin Guo, Shulei Jia

**Affiliations:** 1Jiangsu Key Laboratory of Marine Bioresources and Environment, School of Ocean Food and Biological Engineering, Co-Innovation Center of Jiangsu Marine Bio-Industry Technology, Jiangsu Key Laboratory of Marine Biotechnology, Jiangsu Marine Resources Development Research Institute, Jiangsu Ocean University, Lianyungang 222005, China; t1517027229@163.com (T.T.); ygh19848313141@126.com (G.Y.); 13888923044@163.com (Y.W.); 13770702113@163.com (K.G.); 2School of Basic Medical Sciences, Tianjin Medical University, Tianjin 300070, China; sunsiqi6767@163.com; 3College of Marine Life Science, Ocean University of China, Qingdao 266100, China; 19863605718@163.com

**Keywords:** carnitine–γ-butyrobetaine (γBB)–trimethylamine N-oxide (TMAO) pathway, metabolic gene cluster (MGC), database, cardiovascular diseases

## Abstract

Dietary l-carnitine produces γ-butylbetaine (γBB) in a gut-microbiota-dependent manner in humans, and has been proven to be an intermediate product possibly associated with incident cardiovascular diseases or major adverse events. Eliminating or reducing the production of microbiota-dependent γBB may contribute to adjuvant therapy for cardiovascular diseases. However, to date, our understanding of the γBB metabolic gene clusters (MGCs) and associated microorganisms remains limited. To solve this problem, we constructed a manually curated γBB metabolic gene cluster database (γBMGC) based on Hidden Markov Models (HMMs). It comprised 171,510 allelic genes from 85 species and 20 genera, which could effectively provide high-resolution analysis at the strain level. For simulated gene datasets, with a 50% identity cutoff, we achieved an annotation accuracy, PPV, specificity, F1-score, and NPV of 99.4%, 97.97%, 99.16%, 98.97%, and 100%, respectively, which significantly outperformed existing databases such as KEGG at similar thresholds. The γBMGC database is more accurate, comprehensive, and faster for profiling cardiovascular disease (CVD)-associated genes at the species or strain level, offering a higher resolution in identifying strain-specific γBB metabolic pathways compared to existing databases like KEGG or COG. Meanwhile, we validated the excellent performance of γBMGC in gene abundance analysis and bacterial species distinction. γBMGC is a powerful database for enhancing our understanding of the microbial l-carnitine pathway in the human gut, enabling rapid and high-accuracy analyses of the associated cardiovascular disease processes.

## 1. Introduction

Carnitine, an abundant nutrient in red meat, accelerates the development of atherosclerosis by synthesizing trimethylamine (TMA) and trimethylamine N-oxide (TMAO) through a multistep pathway involving the atherogenic intermediate γ-butylbetaine (γBB). Research has demonstrated the contribution of the gut microbiota to the initial two steps in the meta organismal l-carnitine→γBB→TMA→TMAO pathway in subjects, with γBB being identified as the transitional metabolite [1,2]. Similar to the results previously observed for the supplementation of choline and L-carnitine, the supplementation of γBB also increased the area of atherosclerotic plaque in a manner dependent on gut microbiota activity, particularly involving species like *Escherichia fergusonii* and *Proteus penneri* [2,3]. From a mechanism perspective, the data showed that γBB was metabolized to TMA/TMAO intestinal microorganisms, rather than γBB itself, which was related to the enhancement of atherosclerosis. Various gut microbiota, such as *E. fergusonii*, *Edwardsiella tarda*, and *P. penneri*, can convert dietary carnitine into γBB. In particular, in the human gut, *Emergencia timonensis* bacteria are primarily responsible for the final conversion of γBB into TMA/TMAO, and the abundance of *E. timonensis* microbiota is also influenced by γBB levels [2,3]. This was evidenced by a 15-fold increase in *E. timonensis* abundance when *P. penneri* simultaneously colonized the intestines of mice. Furthermore, in the presence of γBB, the conversion of γBB into TMA/TMAO intensified, implying the complexity of the “meta-metabolome” [3].

Currently, it is known that continuous symbiotic microbial community host interaction is involved in the overall metabolic pathway that affects the development of atherosclerosis, with complex phenotypic characteristics. In this case, it is necessary to investigate the initial steps of the carnitine pathway. Although TMAO had been the focus of previous studies on gut-microbiota-dependent metabolites and atherosclerosis, an association between γBB and human atherosclerosis has also been reported. Studies have indicated that γBB and trimethyllysine (TML), rather than TMAO, are independently associated with cardiovascular mortality [4]. This finding suggests that the pathways related to carnitine metabolism, including γBB, might contribute to carotid atherosclerosis, at least partially independently of TMAO formation [4]. This association is likely to involve both the microbiota-dependent and endogenous pathways, with γBB serving as the most sensitive marker. In addition to its pro-atherogenic effects through TMAO, γBB may directly increase atherosclerosis, thereby illustrating the complex association between carnitine-related metabolites and cardiovascular diseases (CVDs).

To enhance the understanding of γBB synthesis, it is essential to conduct a comprehensive investigation of the bacterial strains that are capable of producing γBB. It has been confirmed that the bacterial synthetic gene cluster *caiTABCDE* is responsible for the synthesis of γBB, which includes the *caiT*, *caiA*, *caiB*, *caiC*, *caiD*, and *caiE* genes [5]. The bacterial *caiTABCDE* gene cluster plays a crucial role in the metabolism of l-carnitine, which may be important to cardiovascular mortality [4]. To date, the MGCs for TMA are the most extensively studied, however, existing research has overlooked the study of the precursors for TMA, such as γBB, which has a positive relationship with an increased concentration of TMA due to an increase in this intermediate metabolite. In this case, conducting long-term studies on γBB as a potential cardiovascular risk marker, analyzing its involvement in the microbiota-dependent carnitine–TMA–TMAO pathway, and exploring the mechanism of γBB as an atherosclerotic mediator are necessary. Previous studies have shown that an association between carnitine metabolites, especially γBB, is associated with carotid atherosclerosis, which may involve a microbiome-dependent mechanism [4]. However, our understanding of the γ-butyrobetaine metabolic microbial communities and their functioning is still limited [1,3,6], which is not conducive to a complete diagnosis in the field of precision medicine.

Currently, several homologous databases, such as KEGG, COG, and SwissProt, are available to decipher functional genes/pathways from metagenomic sequencing data [7,8,9]. These available databases contain a wide variety of genes involved in many metabolic pathways, but still face significant challenges, which include incomplete or unreliable information in public databases, long run times, and a lack of integration and standardization between different resources. Despite the vast scale of biological information accumulated in these databases, providing useful information for research has proven to be more challenging than expected. For example, the CaiT and CaiE proteins have confused and ambiguous annotations in the COG database. The CaiD protein in the KEGG database shares the same KO number (K08299) with other proteins, which is also a chaotic annotation. In addition, most of the available databases lack more potential reference species or stains with an intact *caiTABCDE* gene cluster, the analysis of which is limited to certain species. As quantitative polymerase chain reaction (qPCR) and metagenomic analyses increasingly reveal functional genes in the human gut [10], the recovery of genetic diversity from metagenomic sequencing data has emerged as a significant challenge. Especially at the species or even strain level, high-resolution metagenomic analysis can help to clarify the mechanisms between diversified bacteria and CVDs. Therefore, it is necessary to establish a comprehensive, accurately annotated database for supplementary biomarkers in the preventive treatment of cardiovascular diseases. In this study, we built a well-curated γBB metabolic gene cluster database, referred to as the γBMGC database, to enhance data searches and retrieval efficiency, as well as to furnish analytical tools for other researchers. Due to the fact that each gene in the database comes from a reference strain, the γBMGC database can be used to quickly identify disease-associated species or strains. This can be achieved through simple gene sequence alignment or genome annotations, utilizing its extensive strain coverage to identify the most directly associated strains that affect the development of CVDs, which will contribute to targeting strains for the treatment of CVDs in translational research.

## 2. Materials and Methods

### 2.1. Construction of the γBMGC Database

We developed a modified method for constructing the database, which involved integrating the NCBI RefSeq (ftp://ftp.ncbi.nlm.nih.gov/refseq/release/bacteria/ (accessed on 11 August 2023)) and HumGut (http://arken.nmbu.no/~larssn/humgut/ (accessed on 11 August 2023)) databases (comprising ~116,500 bacterial genomes from the NCBI RefSeq databas). The initial collection of the γBB biosynthetic gene families (*caiTABCDE*) and their functional descriptions were obtained from previous research sources [5,11], then verified in the KEGG and SwissProt databases. KEGG is a valuable and comprehensive database resource that facilitates the analysis of gene functions and provides valuable insights into the utilities of biological systems [8]. Therefore, we refer to the KEGG database for the l-carnitine pathway for the γBB gene family of microbial metabolic processes and functional descriptions. Then, the reference sequences were trained into different Hidden Markov Models (HMMs) using HMMER-build of HMMER v3.3.2 [12].

Next, based on the trained HMMs, we expanded the *caiTABCDE* gene clusters in ~168,484 bacterial genomes with HMMER v3.3.2 (*E*-value: 10^−5^). The homologs were identified, extracted, and integrated by meticulously reviewing the annotation results in the comparison table. For example, homologs with a sequence coverage of less than 80% were filtered out [13]. Then, the obtained results were further processed with HMMER-Extractor (http://82.156.31.238:8000/index (accessed on 10 October 2023)) [14]. The criteria for inclusion and exclusion were as follows: the intact metabolic gene cluster (MGC) of γBB should contain the six genes (*caiTABCDE*), and these genes must be from the same bacterial genome. In this case, the obtained strain-level genes should be the members of the intact *caiTABCDE* gene cluster. This pipeline has been proven to effectively reduce ambiguous annotations [14]. In addition, to avoid errors or omissions, we used the gutSMASH 1.0 [15] to directly annotate the MGCs of γBB in the genomes from the NCBI RefSeq and Human Gut databases, as mentioned above. After that, all the results were checked and merged into a core database. In the results generated by gutSMASH and HMMER-Extractor, MGCs with the same genomic origin in the results were considered to be redundant, and only one result was retained, since each genome generally contains one γBB gene cluster. Representative amino acid sequences and nontarget homologs were deduplicated and clustered with a 100% sequence identity through CD-HIT [16]. This threshold was chosen to eliminate redundancy while retaining unique strain-specific genes. Finally, all representative sequences and homologs were combined to construct the γBMGC database.

### 2.2. Simulated Gene Datasets

As NCBI RefSeq is a curated and comprehensive database with complete gene families [17], those sequences with verified protein names were selected for γBMGC validation. To estimate the accuracy of γBMGC at different thresholds, a simulated gene dataset containing 579 validated sequences and 1432 unrelated sequences from NCBI RefSeq was constructed and compared to γBMGC by using DIAMOND v2.1.8 with an *E*-value of 10^−5^. To assess the accuracy of the γBMGC database, we calculated metrics such as accuracy, positive predictive value (PPV), specificity, sensitivity, and negative predictive value (NPV). A high PPV or NPV ensures maximum true positives or minimal false negatives, which is crucial for detecting rare but functionally significant γBB-related genes. The calculation equations are as follows [13]:(1)$$Accuracy=\frac{\;True\;Positives+True\;Negatives}{True\;Positives+False\;Positives+True\;Negatives+False\;Negatives}$$(2)$$Positive\;predict\;value\;(PPV)=\frac{True\;Positives}{True\;Positives+False\;Positives}$$(3)$$Specificity=\frac{True\;Negatives}{True\;Negatives+False\;Positives}$$(4)$$Sensitivity\;(recall)=\frac{True\;Positives}{True\;Positives+False\;Negatives}$$(5)$$F1-score=\frac{2\times Precision\times Recall}{Precision+Recall}$$(6)$$Negative\;predict\;value\;(NPV)=\frac{True\;Negatives}{True\;Negatives+False\;Negatives}$$

### 2.3. Genome Sequence Datasets from a Mock Community

Given the widespread usage of whole-genome sequencing (WGS) and metagenome sorting, these techniques have proven to be valuable for studying the metabolic pathways of individual microorganisms. To further validate the accuracy of γBMGC, we constructed a mock microbial community that contained 10 bacterial genomes randomly selected from the NCBI GenBank. The mock community sequence dataset was searched against the γBMGC database using DIAMOND v2.1.8 with a 10–100% sequence identity (cov.: 80%, *E*-value ≤ 10^−5^).

### 2.4. Metagenome Sequencing Datasets

To test the application of γBMGC in a variety of human intestinal samples [18,19,20], we analyzed the *caiTABCDE* gene cluster in human fecal samples, including ACVD (*n* = 82), heart failure (*n* = 82), and hypertension (*n* = 82). Metagenomic sequencing datasets were downloaded from the NCBI Sequence Read Archive (SRA) (Appendix A). To mitigate potential fluctuations resulting from diverse sequencing strategies, only metagenomic data generated by paired-end sequencing on the Illumina HiSeq platform were included in the selection process.

### 2.5. Gene Annotation and Abundance Analysis

For each metagenome, sickle was used in the paired-end mode with a minimum quality threshold of 20 to perform quality trimming [21]. The resulting high-quality reads were then assembled into species-level genome bins (SGBs) using metaWRAP v1.3 with the multiple k-mer sizes (parament: --k-list 21, 29, 39, 59, 79, 99, 119, 141) strategy [22]. Then, microbial taxonomy was conducted based on the Genome Taxonomy Database (GTDB). Open reading frames (ORFs) were identified through the utilization of Prodigal v2.6.3 [23] and subsequently annotated by searching against γBMGC using DIAMOND. The DIAMOND search was performed with an *E*-value threshold of ≤10^−5^ (option: -p 20). According to previous research, the comparison results were screened with a ≥50% identity and 80% coverage (amino acids), which is a strict threshold for functional gene annotations [14,24,25]. The abundance was calculated for each gene by using BBMap (https://jgi.doe.gov/data-and-tools/software-tools/bbtools/ (accessed on 2 March 2024)). A species abundance analysis was conducted through MetaPhlAn 4 [26].

### 2.6. Statistical and Case Study

All analyses, unless otherwise specified, were performed by using the R 4.4.1. A random forest analysis was used to identify the most crucial filtering parameters for enhancing the accuracy of the γBMGC database based on the comparison results. The Wilcoxon rank sum test was used to calculate significant differences between genes detected in three human fecal samples or in different databases. The *p*-values were adjusted using Tukey’s multiple comparison test through the utilization of GraphPad software (Version Prism 9.5.1(733), CA, USA).

The significant differences between species- and genus-level metagenomes were analyzed on LEfSe (https://www.omicstudio.cn/tool/60 (accessed on 12 March 2024)). The LEfSe threshold for logarithmic LDA scores for discriminatory characteristics was set at 3.0, and the multiclass analysis strategy was more tightly constrained. All heatmaps were generated on the ImageGP website (https://www.bic.ac.cn/BIC/ (accessed on 2 October 2024)).

## 3. Results

### 3.1. Construction of the Seed Dataset

Since current public anthology databases still have limitations in profiling the γ-butylbetaine microbial community, the γBMGC database was manually developed in this study. Firstly, the reference *caiTABCDE* gene clusters were extracted from the SwissProt and NCBI databases and manually annotated by a keyword search based on protein names or functional descriptions (Appendix A). For those gene families whose sequences were not indexed by SwissProt or NCBI, we performed manual searches in the genomes of certain strains based on 50 studies. The bacterial strains were reported to have the ability to synthetize γ-butylbetaine, so their genomes were downloaded for analysis. Then, we made annotations for the MGCs with gutSMASH, which was conducted based on Hidden Markov Models with a high accuracy. Afterwards, the candidate *caiTABCDE* gene clusters were scrutinized against the annotations to ensure the reliability of the developed database. At last, we constructed a seed dataset comprising of 188 proteins from 22 bacterial species. Furthermore, the seed dataset was trained into six HMMs (CaiT, CaiA, CaiB, CaiC, CaiD, and CaiE) with HMMER v3.3.2 for subsequent analysis (Figure 1a).

### 3.2. The Database of Metabolic Gene Clusters Associated with γ-Butylbetaine

Based on the seed dataset, we made a thorough search against ~168,484 genomes for the homologous genes (Figure 1a). After manually checking and filtering the homologs, ~1566 unqualified genes were filtered, and we finally obtained 171,510 allelic genes, with each of them containing a specific strain origin (Figure 1b) (Appendix A). This is significantly more than the 56 orthologs annotated in KEGG for the same metabolic pathway. Furthermore, with a 100% sequence identity cutoff, 10,734 non-redundant γBB-associated metabolic genes were generated (Figure 1b) (Appendix A). All the genes included nucleotide sequences and amino acids, ensuring functional gene annotations or abundance analysis at the strain level. Notably, each of the obtained bacterial genomes contained at least one *caiTABCDE* gene cluster composed of six metabolic genes in collinearity, indicating that they worked together for the synthesis of γ-butylbetaine. Finally, the constructed database was validated and applied in the metagenome analysis (Figure 1b). When conducting the metagenomic analysis, based on the γBMGC database, we could obtain the gene abundance (metabolic pathway) differences between healthy controls and the disease groups. Meanwhile, with MetaPhlAn 4, the disease-associated species could be further revealed, which could be combined with the gene abundance analysis.

In addition to the reported species, we discovered many other bacterial species with an intact *caiTABCDE* gene cluster. There were, in total, 28,543 bacterial strains from 85 species and 20 genera, which meant that they had the potential to metabolize γ-butylbetaine (Figure 2; Appendix A). To date, these are the most complete statistics for bacterial strains with a possible γBB synthesis capability. The most abundant species was *Escherichia coli* (*n* = 22,330), followed by *Salmonella enterica* (*n* = 3455) (Figure 2a). The most abundant genera were the *Escherichia* and *Salmonella* genera (Figure 2b). The reported species *E. fergusonii* and *P. penneri* were also involved in the amplification of γBMGC (Appendix A). Both species produced TMA and consumed l-carnitine together with γBB, suggesting that the usage of l-carnitine was associated with the presence of *caiTABCDE* genes [2].

### 3.3. Benchmark of the γBMGC Database

To assess the accuracy of the γBMGC database, we initially used a simulated gene dataset. Previous reports have indicated that hit length is not an effective filter for improving database accuracy [13], so we set the default coverage of sequence comparison to 80%, which is the reference threshold for sequence comparison. The fitted model, which demonstrated a high accuracy, indicated that identity was the most crucial factor to distinguish true negatives from false positives (Appendix A). In particular, the PPV value and specificity exhibited substantial improvements as identity increased (coverage ≥ 80%), implying that increasing identity could effectively eliminate false positives (Figure 3a,b). Higher identity thresholds (e.g., 70%) reduced false positives and excluded sequences with natural variability, as seen in Figure 3c. As an illustration, with a 30.0% identity cut-off, where the accuracy was 51.86%, the corresponding PPV and specificity were 37.43% and 32.4%, respectively. Similarly, at a 50% identity cut-off, where accuracy reached 99.4%, the PPV and specificity were 97.97% and 99.16%, respectively. The F1-scores were 54.47% and 98.97% at the 30% and 50% identity cutoffs, respectively (Figure 3c). However, increasing the identity from 20% to 99% did not result in any reduction in sensitivity (Figure 3d). Furthermore, using the aforementioned criteria (50% identity and 80% coverage), the γBMGC database displayed a 100% detection rate for specific *caiTABCDE* genes (Appendix A). Thus, although the sensitivity showed no fluctuations, the 50% identity and 80% coverage criteria exhibited a higher accuracy, PPV, specificity, and F1-score than any other criteria such as the 30.0% identity cut-off, which was a suitable criterion for the detection of the *caiTABCDE* genes.

### 3.4. Validation of γBMGC with a Mock Community

Additionally, we performed a validation of γBMGC by using a mock community consisting of 10 microbial genomes. The results of this validation revealed that all genomes could be detected when using an identity cutoff of 50% (Figure 4a). However, the number of genomes exactly detected was less than 10 genomes when the identity was greater than 70% (Figure 4b). Contrary to the plateau phase observed in the calculation of PPV using the simulated gene dataset, the detection of the *caiTABCDE* gene clusters in certain bacterial genomes (e.g., *Enterobacter lignolyticus* SCF1) exhibited a significant reduction at high identity cutoffs, resulting in false negatives (Appendix A). Among the 10 genomes examined, the number of accurately detected genomes reached a turning point at a 70% identity cut-off point, with detection ratios ranging from 100% to 90%. However, when the sequence identity reached 100%, the detection ratio decreased sharply to 10% (Figure 4b; Appendix A). The chart highlights a sharp drop in detection rates at identity thresholds above 70%, underscoring the importance of balancing sensitivity and specificity (Figure 4b). Therefore, a maximum identity threshold of 70% was considered to be suitable for genome annotation. Using this threshold, the number of accurately estimated genomes was 10, and the accuracy, PPV, F1-score, specificity, and NPV for the simulated gene dataset were 97.91%, 100%, 96.24%, 100%, and 97.15%, respectively (Appendix A).

### 3.5. Performance Comparison Among Different Orthology Databases

To evaluate the performance of the γBMGC database, we compared the accuracy of the γBMGC database with other orthology databases (Figure 4c). The results showed that the γBMGC database contained the most comprehensive gene sets involved in the γ-butyrobetaine metabolic process, with 504 orthology groups, while there were 41, 56, and 58 orthology groups in the COG, KEGG, and SwissProt databases, respectively (Appendix A). As the identity threshold increased, the γBMGC database still retrieved a high percentage of *caiTABCDE* genes, while the other three databases exhibited varying degrees of low detection rates (Figure 4c). For example, under the condition of more than a 90% sequence identity, the performances of different databases began to show differences, especially when the sequence identity reached 100%, and the γBMGC database still showed a high species specificity (93.58%), while the detection rate of the COG, KEGG, and SwissProt databases decreased to 42.78%, 47.06%, and 92.51%, respectively (Appendix A). In this case, we have strong reasons to believe that as test data increase, this proportion will also significantly expand. We also compared the runtime of gene datasets (187 real proteins) using these databases. Among them, the run time of γBMGC (220.48 s) was shorter than that of the COG (781.7 s), KEGG (1831.4 s), and SwissProt (222.14 s) databases (Appendix A). Furthermore, the γBMGC database used the lowest RAM ratio (0%) among all databases (Appendix A). Thus, compared to these existing homology databases, the species-specific γBMGC database allows for a more comprehensive, accurate, and rapid analysis of the *caiTABCDE* gene cluster from sequencing datasets.

### 3.6. Application of γBMGC in Analysis of Cardiovascular Diseases

The developed γBMGC was applied to profile the abundance of *caiTABCDE* genes and its taxonomic abundance using shotgun metagenome sequencing data from the human gut, including atherosclerotic cardiovascular disease (ACVD), heart failure (HF), and hypertension. As determined by the Wilcoxon rank-sum test, after multiple comparison adjustments, the results showed that the abundance of genes involved in the γ-butyrobetaine metabolic process was significantly (*p* < 0.05) different among the three human gut samples (Figure 5a). There was a higher gene abundance in the ACVD, HF, and hypertension samples than in the healthy controls, which showed a significant difference. This meant that the development of the three diseases was possibly associated with the metabolites of γBB in the l-carnitine pathway. As reported, l-carnitine indeed accelerates cardiovascular disease risk through the formation of TMA and TMAO dependent on the gut microbiota through a multistep pathway that involves an intermediate, γ-butyrobetaine [1].

In terms of species, the abundance of *Escherichia coli* was higher in ACVD and hypertension than in the healthy controls, while *Citrobacter analonaticus* was higher in HF than in the healthy controls (Figure 5b). Furthermore, we searched the γBB genes in these SGBs, and detected an intact *caiTABCDE* gene cluster in these genomes. Similarly, at the genus level, the species significantly enriched in ACVD, HF, and hypertension were *Escherichia*, *Citrobacter*, and *Escherichia*, respectively, followed by the genera *Pseudocitrobacter*, *Yokenella*, and *Shewanella*, respectively (Figure 5b). Notably, *Citrobacter* spp. was consistently enriched across the ACVD, HF, and hypertension samples. Obviously, metagenomic analysis through the γBMGC database showed a more comprehensive result than the existing studies [1,2]. It is worth mentioning that the search time for γBMGC was 3837.2–4604.8 s, which was less than that of COG (7600.4–9200.6 s), SwissProt (3930.2–4800 s), and KEGG (30312.3–36370.9 s). Thus, these results suggest that the γBMGC database can comprehensively, accurately, specifically and rapidly analyze the microbial communities associated with γBB metabolism.

## 4. Discussion

The bacterial *caiTABCDE* gene cluster is critical for the dietary transition of l-carnitine to γBB and TMAO in the host and contributes to the risk of cardiovascular disease [1]. As shown in Appendix A, species with intact *caiTABCDE* clusters were predominantly associated with increased γBB levels. It is important to rapidly decipher human γBB metabolism genes from metagenomic sequencing data using precise databases. In this study, we developed the γBMGC database containing six gene families, identified key criteria (i.e., identity) to ensure its annotation accuracy, and used it to analyze the *caiTABCDE* gene cluster from multiple different human intestinal samples. The results indicate that γBMGC is a powerful database that can accurately, comprehensively, and rapidly annotate the *caiTABCDE* gene from the human gut.

Compared to other databases, the γBMGC database offers more accurate annotations for metagenomic sequencing data. Firstly, γBMGC provides an improved annotation of the *caiTABCDE* gene families. In the COG database, certain enzymes are poorly annotated, leading to ambiguities. For example, the gene encoding the L-carnitine/gamma-butyrobetaine antiporter (referred to as *caiT* in this study) and the gene encoding the carnitine abrogating enzyme (referred to as *caiE* in this study) are labeled “transporter (NP_308070.1)” and “phenylacetic acid degradation protein PaaY (NP_459074.1)” in COG, respectively, causing confusion and ambiguity. Moreover, in the KEGG database, the genes for *caiD* (carnitine-CoA_dehydrogenase) and a putative protein (cko: CKO_03347) share the same KO number (K08299), further complicating the annotation process. Secondly, γBMGC significantly reduces false positives by incorporating 10,734 nonredundant homologs and implementing optimal filtering parameters. When aiming to obtain functional or taxonomic annotations from metagenomic sequencing datasets, researchers commonly search for query sequences in the NCBI RefSeq database, which comprises a comprehensive set of standard sequences with various functions [17]. The bit score and the *E*-value are widely used filtering criteria to enhance the accuracy of functional annotations [27]. Although the homologous method helps to exclude some non-γBB gene operons, false positives may still arise when non-γBB sequences exhibit a higher identity with the *caiTABCDE* genes compared to their homologs. This represents the most common error encountered when directly utilizing alignment results for downstream functional analysis. In previous studies, functional databases such as the integrase and CARD databases were typically limited to a high identity cutoff of ≥80% for prediction [28,29]. More recently, nitrogen cycling gene annotation was performed using NCycDB with an identity threshold of 85% [30]. However, it is important to note that setting a high cutoff can lead to an increase in false negatives, meaning that a significant proportion of genuine functional genes may be excluded [31]. Thus, selecting an appropriate cutoff point is crucial to effectively reduce both false positives and false negatives. In this study, the random forest analysis highlighted that identity, when combined with a certain coverage, emerged as the most influential screening factor for enhancing annotation accuracy. With a 50% identity (80% coverage) cut-off, the accuracy, PPV, specificity, and NPV were 99.4%, 97.97%, 99.16%, and 100%, respectively. Using the combination of a 70% sequence identity and 80% coverage, all performance metrics, including accuracy, PPV, specificity, and NPV, exceeded 96%. This indicates that γBMGC exhibits a high accuracy in profiling the *caiTABCDE* gene cluster within the sequencing dataset. Empirically, we set the hit length threshold at 80%, as it effectively annotates the majority of predicted ORFs without a significant increase in false negatives. It is worth noting that, under a 100% identity cutoff, some genomes within the simulated community exhibited detection rates below 10%. A strain, specifically *Enterobacter lignolyticus* SCF1, would still be missed, even with a 90% identity cutoff. Therefore, we recommend using a 50% identity cutoff combined with 80% coverage to analyze the *caiTABCDE* gene cluster in the metagenomic sequencing data. This approach ensures the detection of all known genes while maintaining a low false-positive rate (0.84%). Alternatively, a more stringent cut-off of 70.0% identity can be employed to identify the *caiTABCDE* gene operons from genomes, which would further minimize false positives. Another advantage of γBMGC is that it contains a more extensive collection of representative *caiTABCDE* gene sequences compared to other databases. COG, SwissProt, and KEGG have fewer representative sequences of the *caiTABCDE* gene, particularly the *caiA* gene, potentially compromising the diversity and abundance of certain genes in those databases.

To understand the γBB metabolic microbial communities in human CVD-related diseases, the profile of *caiTABCDE* genes was deciphered using γBMGC. The results revealed that ACVD, HF, and hypertension were possibly associated with intermediate γ-butyrobetaine metabolites (*p* < 0.05), which was consistent with the existing research. For example, it was reported that there were elevated levels of γBB and carnitine in patients with heart failure [32], and more recently, γBB and carnitine were found to be strongly associated with clinical atherosclerosis [4]. Although γBB may not be a direct metabolite causing cardiovascular diseases, an increase in γBB is often accompanied by an increase in TMA conversion efficiency due to the coexistence of “meta-metabolome” [33,34]. Therefore, controlling the microbial community that synthesizes γBB can help to slow down the occurrence of cardiovascular diseases. Based on the analysis of the species abundance of γBB, it is possible to interpret the species that may be associated with cardiovascular disease. In ACVD, the abundance of Enterobacteriaceae bacteria, including *Escherichia coli*, *Klebsiella* spp., and *Enterobacter aerogenes*, was higher than that in the healthy controls [18]. Our results demonstrated that *Escherichia coli* with intact *caiTABCDE* gene clusters was, indeed, enriched in the disease samples, while the other species might be associated with other metabolites. Similarly, in HF and hypertension, the enriched species were the *Citrobacter*, *Yokenella* genus and the *Escherichia*, *Shewanella* genus, respectively, which further supplements existing research [19,20]. Our findings revealed an interrelationship between γBB, red meat intake, and cardiovascular disease risk, providing evidence at the level of specific genes for study of the l-carnitine metabolic pathway, such as the enrichment of genes from *E. coli* in ACVD samples (Figure 5b). The results of gene abundance and classification demonstrated that γBMGC was a sensitive, accurate, and broad-spectrum database for analyzing the *caiTABCDE* gene clusters in the human gut.

It is worth mentioning that, as an intermediate metabolite, although γBB cannot directly affect cardiovascular disease, its elevation is often accompanied by an increase in TMA, which is a widely reported bacterial metabolite related to CVDs. Therefore, in addition to the γBB pathway, the γBMGC database should also be applied to the TMA metabolic pathway. In particular, many bacterial strains generally have the ability to synthesize both γ-butylbetaine and TMA. In this case, this database can be used to supplement the previously established TMA database [35] for a more comprehensive analysis of CVD-associated microbial communities. However, there are still limitations in this study. For example, the bacterial genomes lacked timeliness, and the number of CVD-associated samples was too small. The reliance on existing databases for initial sequence collection usually introduces potential biases. Thus, in future updates of γBMGC, it is not possible to directly collect gene sequences from other databases, but reliable homologous genes should be obtained by setting reasonable thresholds and using existing tools for sequence alignment from microbial genomes. In addition, with an increase in studies, more validated species should be included as references for training HMMs to obtain a more comprehensive and extensive gene dataset of γBB. This is due to the rigor of the database construction process, and typically, searching for intact gene clusters can effectively ensure a reduction in false positives. The construction process of the γBMGC database used reference genomes, which are more representative than the draft genomes (SGBs) assembled from metagenomes. It underwent a rigorous screening process to ensure that each gene originated from a specific bacterial strain. Furthermore, the γBMGC database showed high accuracy in both a simulated gene dataset and mock community for validation.

## 5. Conclusions

In this study, we developed a highly accurate and comprehensive functional gene database (γBMGC) with metrics such as a 99.4% accuracy and 97.97% PPV, validating its robustness. It was specifically designed for assisting the analysis of cardiovascular diseases, which could classify CVD-associated genes at the strain level. To enhance the accuracy, an adaptive threshold was applied to microbial communities within the human gut, and it was proven to be accurate in distinguishing the *caiTABCDE* gene cluster at a high resolution. Therefore, the γBMGC database serves as a powerful tool for efficiently analyzing γBB microbial communities and understanding their underlying mechanisms within CVDs. Under the “One Health” concept, it could serve as a valuable reference for the targeted diagnosis and comprehensive treatment of emerging CVDs.

## Figures and Tables

**Figure 1 microorganisms-13-00225-f001:**
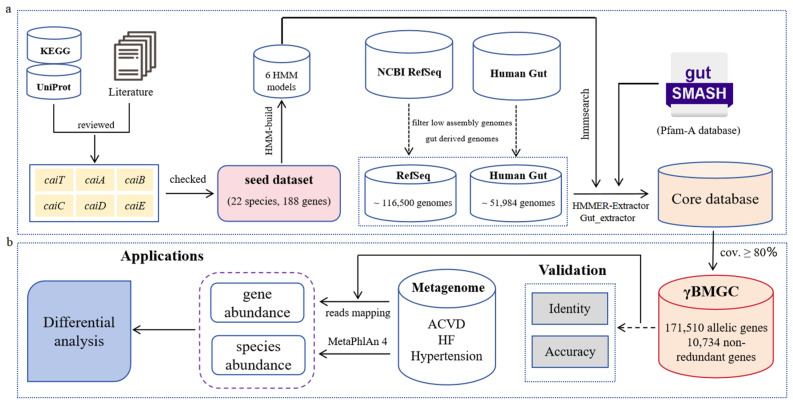
The technical flowchart for the γBMGC database construction. The flowchart includes data collection, sequence search (**a**), database validation, and applications in metagenome (**b**). The homologs were searched in ~168,484 genomes with HMMER and gutSMASH. Then, the sequences were extracted through the HMMER-Extractor and Gut_extractor scripts, which were implemented with Python. Finally, all genes were checked and curated for constructing the γBMGC database. The best identity and accuracy test of γBMGC were conducted in a mock community and simulated gene dataset, respectively.

**Figure 2 microorganisms-13-00225-f002:**
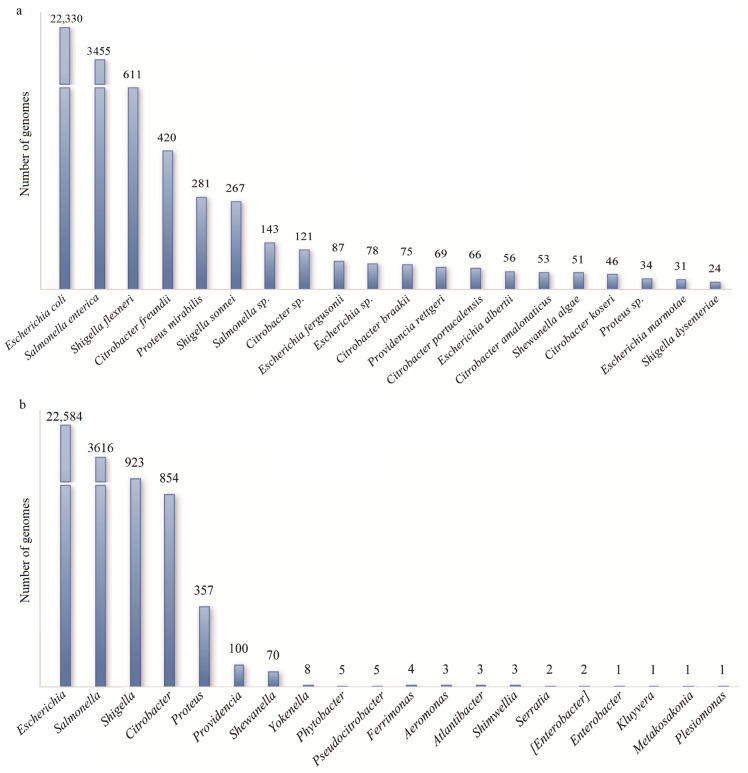
Statics of strains at species (**a**) and genus level (**b**) in the γBMGC database. At species level, statistical analysis was conducted based on the top 20 bacterial species.

**Figure 3 microorganisms-13-00225-f003:**
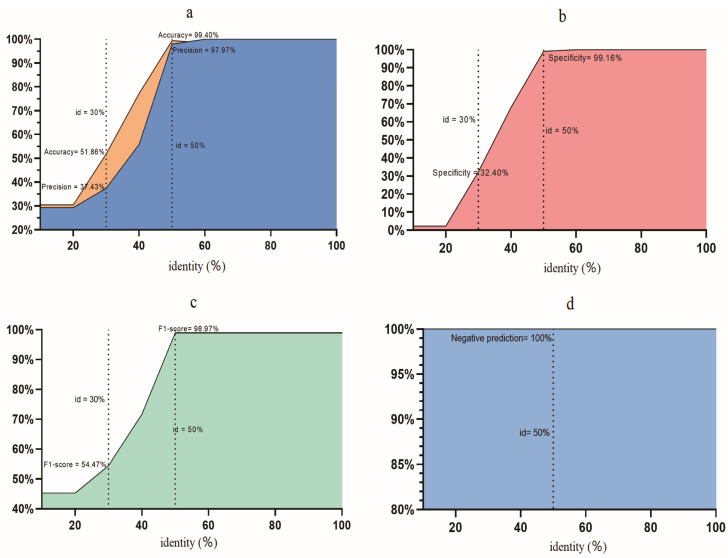
The accuracy of γBMGC against sequence identity (coverage 80%). The accuracy and positive predictive value (**a**), specificity (**b**), F1-score (**c**), and negative predictive value (**d**) were recorded along with the identity varied from 0 to 100% with a step by 10%. Left dash line represents a 30% identity cutoff, and right dash line means a 50% identity cutoff.

**Figure 4 microorganisms-13-00225-f004:**
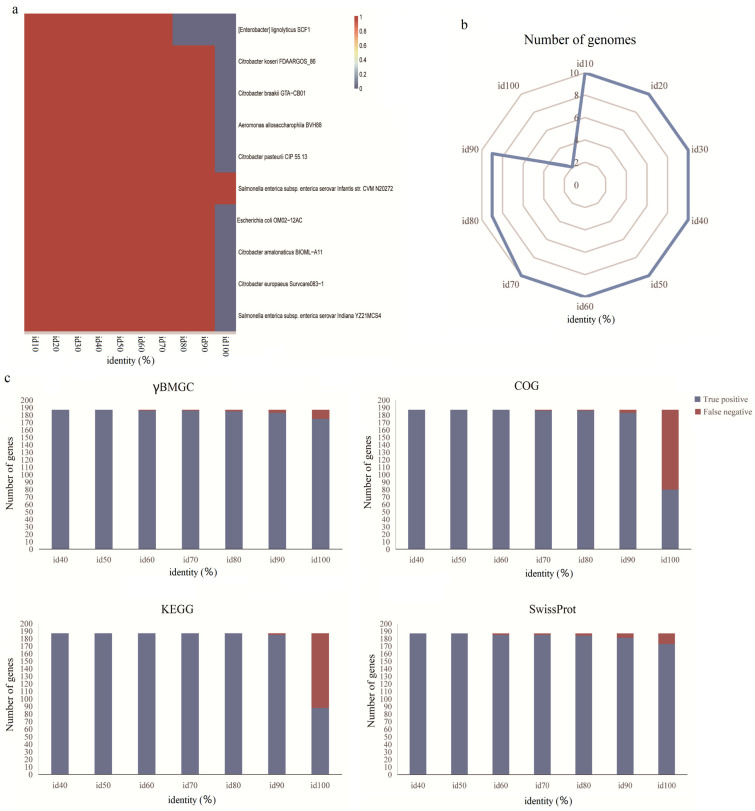
The accuracy and completeness of γBMGC validated by a mock community. (**a**) Heatmap showed the detection of each genome involved in the mock community as identity increased, and red represents the perfect detection. (**b**) Radar chart showed the number of genomes detected at different identity cutoffs. (**c**) Detection comparison among different databases.

**Figure 5 microorganisms-13-00225-f005:**
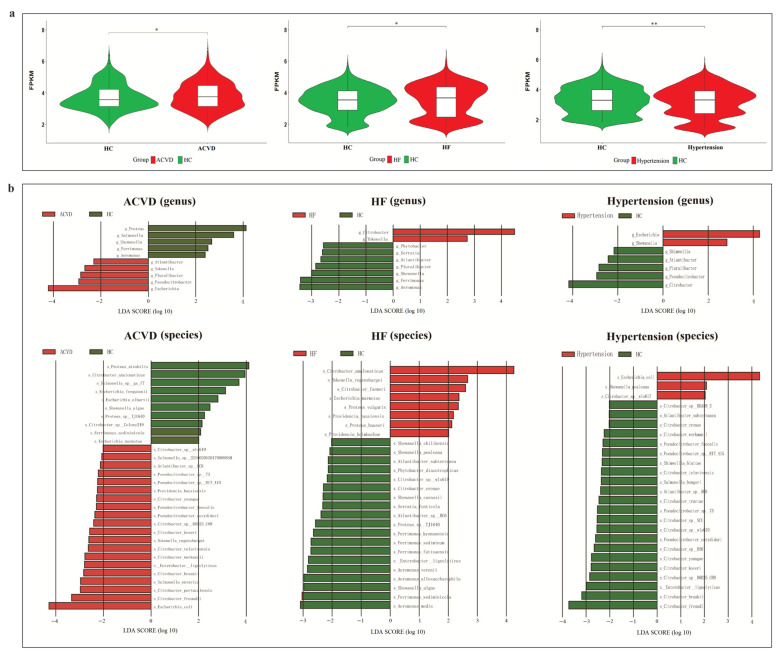
Abundance comparison between healthy controls (HCs) and the disease group. (**a**). Gene abundance and (**b**). species and genus abundance. ACVD: atherosclerotic cardiovascular disease; HF: heart failure. Comparison of gene abundance among different samples is based on rank sum test (** marks *p* value < 0.01, * means *p* value < 0.05).

## Data Availability

The γBMGC database is available at https://github.com/xielisos567/caiTABCDE (accessed on 1 January 2025). This repository includes curated gene datasets, validation scripts, and the γBMGC database files. The metagenomic shotgun-sequencing data for all ACVD samples have been deposited in the European Bioinformatics Institute (EBI) database under the accession code ERP023788. The dataset for hypertension was deposited in the EMBL European Nucleotide Archive (ENA) under BioProject accession code PRJEB13870. The dataset for chronic heart failure (CHF) was requested from the corresponding author (J.Z., email: fwzhangjian62@126.com; J.C., email: caijun@fuwaihospital.org). The ethical approvals and data-sharing permissions for the metagenomic datasets used in this study could be checked in the published articles accordingly.

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
