# Peer review of "γBMGC: A Comprehensive and Accurate Database for Screening TMAO-Associated Cardiovascular Diseases"

_microorganisms, 2025, doi:10.3390/microorganisms13020225_

Round 1
Reviewer 1 Report
Comments and Suggestions for Authors
microorganisms-3420429-peer-review-v1
Dear Editor
The current work looks very interesting and represent an complex research project. The research project is based on bioinformatics and appropriate analysis of existing data bases. Personally, I admire bioinformatics and have high respect for such studies. My self I am not bioinformatic and in several parts of the present work will not be the best option to judge the current study. From the point of view of the microbiologist, the present work looks like well-planned and performed, however, opinion of specialist in area of bioinformatic will be more than appropriate for this kind of study.
One of the question is, however, how this study fits to the scope of the journal as Microorganisms. In my opinion this paper will be more appropriate for the bioinformatic or medical related journal, since the focus of the manuscript fits much more to these two areas. However, if the Editor of the journal believe that current manuscript is appropriate to Microorganisms, then I will accept his decision.
According to my understanding and knowledge (I will again underline that my expertise is in classical and biomolecular microbiology, but not bioinformatics), paper looks like a well-structured, with balance between different parts, informative introduction, material and methods with sufficient details; Results and further discussion well-structured and on the focus of the aim of the study.
References are out of the standards from the publisher and Microorganisms, and this needs to be adjusted by the authors.
Something looks wrong with the authors list. The authors list ends with "and". This is a missing author. Or do authors want to say that S. Jia is affiliated with "b" and is corresponding author? Please, correct it.
Ln49-50: This sentence is missing something. Please, check and complete it.
Author Response
Many thanks for your comments. In this study, γ-butyrobetaine (γBB) was mentioned as an important microbial intermediate metabolite. Based on the metabolite, we established a curated γBB synthesis gene cluster database, laying the foundation for the development of biomedical science. Therefore, In my opinion this paper fits to the scope of the journal as Microorganisms (Special issue: Secondary Metabolism of Microorganisms). Revisions are made to the manuscript in red color. Thanks for your time and effort in helping us improve the manuscript and look forward to your final decision on publication.
- References are out of the standards from the publisher and Microorganisms, and this needs to be adjusted by the authors.
Response: Thanks for your suggestion. Modifications made and every reference cited in the text is also present in the reference list with reference style of Microorganisms.
- Something looks wrong with the authors list. The authors list ends with "and". This is a missing author. Or do authors want to say that S. Jia is affiliated with "b" and is corresponding author? Please, correct it.
Response: Thanks for your suggestion. Modifications made.
- Ln49-50: This sentence is missing something. Please, check and complete it.
Response: Sorry for our carelessness. According to your suggestions, we have checked and completed this sentence. For the details, see the text, please.

Reviewer 2 Report
Comments and Suggestions for Authors
The manuscript introduces the γBMGC database, a novel and specialized tool for profiling γ-butyrobetaine (γBB) metabolic gene clusters. The focus on γBB as an intermediate in the gut microbiota-dependent l-carnitine pathway addresses an important research gap in understanding the microbial contributions to cardiovascular diseases (CVDs). This work highlights the γBMGC database's potential for providing accurate, comprehensive, and efficient annotations of CVD-related microbial pathways, setting it apart from existing orthology databases.
1. The introduction effectively establishes the clinical significance of γBB and its association with atherosclerosis. To enhance the introduction, the authors should provide a more explicit comparison of γBMGC with current databases such as KEGG, COG, and SwissProt. For instance, including specific examples of how these databases fail to comprehensively capture γBB-associated gene clusters would clarify the necessity for developing γBMGC. The authors could also expand on how the database could be used in translational research, particularly its potential applications in diagnostic or therapeutic settings for CVD.
2. The methods section demonstrates strong technical rigor, incorporating diverse datasets and validation metrics. The detailed construction of the γBMGC database through the integration of NCBI RefSeq, HumGut, and Pfam-A datasets provides a robust foundation. However, certain methodological steps lack sufficient detail for full reproducibility. The process of manual curation of gene clusters, including criteria for inclusion and exclusion, should be described in more detail. For example, explaining how ambiguous annotations were resolved or how redundancy was managed would help clarify the process. Additionally, the rationale for choosing specific thresholds, such as 50% identity and 80% coverage for genome annotation, should be expanded with more empirical evidence or comparisons to other potential thresholds.
3. The results effectively validate the database, demonstrating its high accuracy, specificity, and speed compared to existing resources. The authors highlight the superior performance of γBMGC in annotating γBB-related genes from simulated datasets, mock communities, and human metagenomic samples. This validation strongly supports the database’s practical utility. However, the presentation of results could be improved. For instance, Figures 3 and 4 require better resolution and more descriptive legends to enhance interpretability. A flowchart depicting the workflow for applying γBMGC to metagenomic datasets would also help readers understand how the database can be practically utilized.
5. In the discussion, the authors make a convincing case for γBMGC’s relevance to microbial genomics and its contribution to the understanding of CVD pathogenesis. However, some limitations of the study should be addressed more explicitly. The reliance on existing databases for initial sequence collection introduces potential biases, which should be discussed along with strategies to mitigate these issues in future updates of γBMGC. The discussion could also consider the database’s applicability to other metabolic pathways beyond γBB, exploring the potential for broader use cases or the need for additional databases to complement this resource.
6. Ethical considerations surrounding the use of human metagenomic data are only briefly mentioned. The authors should provide explicit details about ethical approvals and data-sharing permissions for the metagenomic datasets used. Addressing these points would strengthen the manuscript’s adherence to research integrity and transparency standards.
7. In conclusion, the γBMGC database represents a valuable addition to the field of microbial genomics, particularly in the context of CVD research. The manuscript successfully demonstrates its utility and performance but could benefit from several refinements. Specifically, the authors should expand the introduction to emphasize the unique advantages of γBMGC, provide greater methodological transparency in describing database construction and validation, and improve the clarity of data presentation. Addressing these points would significantly enhance the impact of the study and establish γBMGC as a cornerstone resource for future research on microbial contributions to human health and disease.
Line 19: The phrase “comprehensive and accurate” could be expanded to explain how these qualities specifically benefit the analysis of γBB metabolism compared to other tools. Consider rephrasing to: “comprehensive and accurate, offering a higher resolution in identifying strain-specific γBB metabolic pathways compared to existing databases like KEGG or COG.”
Line 22: The accuracy metrics mentioned here are impressive, but the context for these metrics is missing. Add a brief comparison, such as: “achieving an annotation accuracy of 99.4%, which significantly outperforms existing databases like KEGG at similar thresholds.”
Line 39: The phrase “gut microbial dependent manner” would benefit from clarification. Suggest replacing it with: “in a manner dependent on gut microbiota activity, particularly involving species like Escherichia fergusonii and Proteus penneri.”
Line 49: The sentence “The conversion of γBB to TMA/TMAO intensified…” introduces an important concept but lacks supporting data. Suggest adding a reference to the relevant figure or supplementary table that confirms this observation.
Line 69: The mention of “bacterial synthetic gene cluster caiTABCDE” would benefit from a diagram or figure reference explaining the cluster’s role in γBB metabolism. Consider suggesting:
“Figure 1 illustrates the organization of the caiTABCDE gene cluster and its metabolic role in γBB synthesis.”
Line 104: The reference to datasets accessed on August 11, 2023, should include additional information about dataset size and composition. Suggest including a parenthetical:
(“comprising ~116,500 bacterial genomes from the NCBI RefSeq database”).
Line 126: The final database was curated using CD-HIT with 100% sequence identity. Explain why this threshold was selected, as stricter thresholds might exclude closely related sequences. Add a justification: “This threshold was chosen to eliminate redundancy while retaining unique strain-specific genes.”
Line 138: In the validation formula section, explicitly describe the practical implications of metrics like PPV or NPV in annotating metagenomic datasets. For instance: “A high NPV ensures minimal false negatives, crucial for detecting rare but functionally significant γBB-related genes.”
Line 186: The mention of 171,510 allelic genes lacks a direct comparison to other databases. Add a comparative statement: “This is significantly more than the 56 orthologs annotated in KEGG for the same metabolic pathway.”
Line 230: When presenting the 50% identity cutoff results, explain why increasing identity thresholds leads to diminishing returns. Suggest adding: “Higher identity thresholds (e.g., 70%) reduced false positives but excluded sequences with natural variability, as seen in Figure 3c.”
Line 256: The radar chart in Figure 4b is referenced but not described in sufficient detail. Suggest adding: “The chart highlights a sharp drop in detection rates at identity thresholds above 70%, underscoring the importance of balancing sensitivity and specificity.”
Line 298: The statement that gene abundance is “significant (p < 0.05)” lacks statistical detail. Add the specific test used, such as: “as determined by the Wilcoxon rank-sum test, adjusted for multiple comparisons.”
Line 309: The term “interesting” to describe Citrobacter spp.’s presence in multiple diseases is imprecise. Replace with: “notable, as Citrobacter spp. was consistently enriched across ACVD, HF, and hypertension samples.”
Line 318: The discussion of caiTABCDE’s role in CVD should reference a specific part of the results. Suggest: “As shown in Table S4, species with intact caiTABCDE clusters were predominantly associated with increased γBB levels.”
Line 376: The phrase “providing evidence at the level of specific genes” would benefit from clarification. Add a brief example: “such as the enrichment of Escherichia coli genes in ACVD samples (Figure 5b).”
Line 397: The conclusion mentions “high accuracy” but does not reiterate key metrics. Consider adding: “with metrics such as 99.4% accuracy and 97.97% PPV validating its robustness.”
Line 408: The GitHub link for data availability is mentioned, but the contents should be briefly described. Suggest: “This repository includes curated gene datasets, validation scripts, and the γBMGC database files.”
Line 428: Add: “All datasets were obtained from publicly accessible repositories with prior ethical approvals, as listed in Table S2.”
Figure 1 (Line 195): The flowchart lacks sufficient annotations. Add labels or notes explaining the key steps in the database construction pipeline, such as sequence extraction and HMM training.
Author Response
We greatly appreciate your efforts and valuable comments that have enhanced our manuscript. We have meticulously revised the manuscript and the English writing, addressing each of your comments in detail. Additionally, we have further refined the article's details, clarity, and depth in the revised manuscript. We sincerely hope that you find these revisions and our responses satisfactory. Please review our responses and the corresponding changes made throughout the manuscript.
- The introduction effectively establishes the clinical significance of γBB and its association with atherosclerosis. To enhance the introduction, the authors should provide a more explicit comparison of γBMGC with current databases such as KEGG, COG, and SwissProt. For instance, including specific examples of how these databases fail to comprehensively capture γBB-associated gene clusters would clarify the necessity for developing γBMGC. The authors could also expand on how the database could be used in translational research, particularly its potential applications in diagnostic or therapeutic settings for CVD.
Response: We thank your constructive comments. We have provided a more explicit comparison of γBMGC with current databases such as KEGG, COG, and SwissProt and the potential applications in diagnostic or therapeutic settings for CVD of the γBMGC database.
The revised text reads as follows on lines 95-100 and 109-114:
“For example, the CaiT and CaiE proteins had confused and ambiguous annotations in the COG database. The CaiD protein in KEGG database shared the same KO number (K08299) with other proteins, which was also a chaotic annotation. In addition, most of the available databases lacked more potential reference species or stains with intact caiTABCDE gene cluster, the analysis of which was limited to certain species.
Due to the fact that each gene in the database comes from a reference strain, the γBMGC database can be used to quickly identify the disease associated species or strains. This can be achieved through simple gene sequence alignment or genome an-notations, utilizing its extensive strain coverage to identify the most directly associated strains that affect the development of CVDs, which will contribute to target the strains for treatment of CVDs in translational researches.”
- The methods section demonstrates strong technical rigor, incorporating diverse datasets and validation metrics. The detailed construction of the γBMGC database through the integration of NCBI RefSeq, HumGut, and Pfam-A datasets provides a robust foundation. However, certain methodological steps lack sufficient detail for full reproducibility. The process of manual curation of gene clusters, including criteria for inclusion and exclusion, should be described in more detail. For example, explaining how ambiguous annotations were resolved or how redundancy was managed would help clarify the process. Additionally, the rationale for choosing specific thresholds, such as 50% identity and 80% coverage for genome annotation, should be expanded with more empirical evidence or comparisons to other potential thresholds.
Response: Thank you for your valuable suggestions regarding the methods section. We appreciate your emphasis on clarity and thoroughness in the description of our methods, and modifications made. In addition, as the rationale for choosing specific thresholds, we mainly refer to existing literature for our research. Sequence similarity network (SSN) analysis is a novel and effective method based on homology function prediction, which links characterized and uncharacterized genes with biochemical functions, and can quickly determine target gene sequences. When conducting functional gene searches, normally, a useful "rule of thumb" is that the sequence identity of the isofunctional families often shares > 40% (E-value < 1E-60) to output the initial SSN so that the isofunctional families cannot be separated, which means minimizing false positive results as much as possible [14, 24]. Meanwhile, previous researches had analyzed and compared reasonable threshold ranges, and found that the 50% threshold could be used as a strict threshold for functional genes search [25]. Thus, we here introduced references as a basis in the methods section and revised the sentences accordingly.
The revised text reads as follows on lines 130-140, 142-144, 146-147, 157-157, 189, 194 and 197:
“Next, based on the trained HMMs, we expanded the caiTABCDE gene clusters in ~ 168,484 bacterial genomes with HMMER v3.3.2 (E-value: 10-5). The homologs were identified, extracted, and integrated by meticulously reviewing the annotation results in the comparison table. For example, the homologs with sequence coverage of less than 80% were filtered out [13]. Then, the obtained results were further processed with HMMER-Extractor (http://82.156.31.238:8000/index) [14]. The criteria for inclusion and exclusion were as follows: the intact metabolic gene cluster (MGC) of γBB should contain the six genes (caiTABCDE), and these genes must be from the same bacterial genome. In this case, the obtained strain-level genes should be the members of the intact caiTABCDE gene cluster. This pipeline has been proved to effectively reduce ambiguous annotations [14]. In addition, to avoid errors or omissions, we used the gutSMASH software [15] to directly annotate the MGCs of γBB in the genomes from NCBI RefSeq and Human Gut databases as mentioned above.
In the results generated by gutSMASH and HMMER-Extractor, the MGCs with the same genomic origin in the results are considered to be redundant, and only one result will be retained since each genome generally contains one γBB gene cluster.
A high PPV or NPV ensures maximum true positives or minimal false negatives, which is crucial for detecting rare but functionally significant γBB-related genes. The calculation equations are as follows [13]:
Then, the microbial taxonomy was conducted based on the Genome Taxonomy Database (GTDB).
According to the previous researches, the comparison results were screened with ≥ 50% identity and 80% coverage (amino acids), which was a strict threshold for functional gene annotations [14, 24-25].
The species abundance analysis was conducted through MetaPhlAn 4 [26].
References:
- J. Zeng, Q. Tu, X. Yu, et al., PCycDB: a comprehensive and accurate database for fast analysis of phosphorus cycling genes. Microbiome. 2022; 10:101. doi: 10.1186/s40168-022-01292-1.
- J. Yang, S. Sun, N. Sun, et al. HMMER-Extractor: an auxiliary toolkit for identifying genomic macromolecular metabolites based on Hidden Markov Models. Int J Biol Macromol. 2024; 283(Pt 2):137666. doi: 10.1016/j.ijbiomac.2024.137666.
- N. Oberg, R. Zallot, J.A. Gerlt. EFI-EST, EFI-GNT, and EFI-CGFP: Enzyme function initiative (EFI) web resource for genomic enzymology tools. J Mol Biol. 2023; 435(14):168018. doi: 10.1016/j.jmb.2023.168018.
- R. Zallot, N. Oberg, J.A. Gerlt. The EFI web resource for genomic enzymology tools: leveraging protein, genome, and metagenome databases to discover novel enzymes and metabolic pathways. Biochemistry. 2019; 58(41):4169-4182. doi: 10.1021/acs.biochem.9b00735.
- P. Manghi, A. Blanco-Míguez, S. Manara, et al. MetaPhlAn 4 profiling of unknown species-level genome bins improves the characterization of diet-associated microbiome changes in mice. Cell Rep. 2023; 42(5):112464. doi: 10.1016/j.celrep.2023.112464.
- The results effectively validate the database, demonstrating its high accuracy, specificity, and speed compared to existing resources. The authors highlight the superior performance of γBMGC in annotating γBB-related genes from simulated datasets, mock communities, and human metagenomic samples. This validation strongly supports the database’s practical utility. However, the presentation of results could be improved. For instance, Figures 3 and 4 require better resolution and more descriptive legends to enhance interpretability. A flowchart depicting the workflow for applying γBMGC to metagenomic datasets would also help readers understand how the database can be practically utilized.
Response: Many thanks for the reviewer's comments. We have redrawn figure 1, figure 3 and figure 4, with high resolution and descriptive legends, which can help readers understand how the database can be practically utilized. For the details, please see the text.
- In the discussion, the authors make a convincing case for γBMGC’s relevance to microbial genomics and its contribution to the understanding of CVD pathogenesis. However, some limitations of the study should be addressed more explicitly. The reliance on existing databases for initial sequence collection introduces potential biases, which should be discussed along with strategies to mitigate these issues in future updates of γBMGC. The discussion could also consider the database’s applicability to other metabolic pathways beyond γBB, exploring the potential for broader use cases or the need for additional databases to complement this resource.
Response: Many thanks for the reviewer's comments. We have added the discussions on this section in the last paragraph as follows on lines 450-472:
“It is worth mentioning that as an intermediate metabolite, although γBB cannot directly affect cardiovascular disease, its elevation is often accompanied by an increase in TMA, which is a widely reported bacterial metabolite related to CVDs. Therefore, in addition to the γBB pathway, the γBMGC database should also be applicability to the TMA metabolic pathway, especially, many bacterial strains generally have the abilities to synthesize both γ-butylbetaine and TMA. In this case, this database can be used to supplement the previously established TMA database [35] for a more comprehensive analysis of the CVD associated microbial communities. However, there were still limitations in this study. For example, the bacterial genomes lacked timeliness, and the number of CVD associated samples was too small. The reliance on existing databases for initial sequence collection usually introduces potential biases. Thus, in future updates of γBMGC, it is not possible to directly collect gene sequences from other databases, but reliable homologous genes should be obtained by setting reasonable thresholds and using existing tools for sequence alignment from microbial genomes. In addition, with the increase of researches, more validated species should be included as references for training the HMMs to obtain a more comprehensive and extensive gene dataset of γBB.”
- Ethical considerations surrounding the use of human metagenomic data are only briefly mentioned. The authors should provide explicit details about ethical approvals and data-sharing permissions for the metagenomic datasets used. Addressing these points would strengthen the manuscript’s adherence to research integrity and transparency standards.
Response: Thanks for pointing this out. We have added descriptions of the associated metagenomic datasets used in the data availability statement and Ethics approval and consent to participate (lines 484-492 and 511-512). As the metagenomic dataset are sourced from published papers, the ethical approvals and data-sharing permissions for the metagenomic datasets used in this study can be accessed in these published articles accordingly in in the revised manuscript [18-20].
Reference:
- Z. Jie, H. Xia, S.L. Zhong, et al. The gut microbiome in atherosclerotic cardiovascular disease. Nat Commun. 2017; 8:845. doi: 10.1038/s41467-017-00900-1.
- X. Cui, L. Ye, J. Li, et al. Metagenomic and metabolomic analyses unveil dysbiosis of gut microbiota in chronic heart failure patients. Sci Rep. 2018; 8:635. doi: 10.1038/s41598-017-18756-2.
- J. Li, F. Zhao, Y. Wang, et al. Gut microbiota dysbiosis contributes to the development of hypertension. Microbiome. 2017; 5:14. doi: 10.1186/s40168-016-0222-x.
- In conclusion, the γBMGC database represents a valuable addition to the field of microbial genomics, particularly in the context of CVD research. The manuscript successfully demonstrates its utility and performance but could benefit from several refinements. Specifically, the authors should expand the introduction to emphasize the unique advantages of γBMGC, provide greater methodological transparency in describing database construction and validation, and improve the clarity of data presentation. Addressing these points would significantly enhance the impact of the study and establish γBMGC as a cornerstone resource for future research on microbial contributions to human health and disease.
Response: Thanks for your valuable suggestions. Modifications made the introduction to emphasize the unique advantages of γBMGC, provide greater methodological transparency in describing database construction and validation, and improve the clarity of data presentation.
The organization of the caiTABCDE gene cluster and its metabolic role in γBB synthesis were illustrated in figure 1 as described below.
The revised text reads as follows on lines 109-114, 132-144, and 197-197:
Due to the fact that each gene in the database comes from a reference strain, the γBMGC database can be used to quickly identify the disease associated species or strains. This can be achieved through simple gene sequence alignment or genome annotations, utilizing its extensive strain coverage to identify the most directly associated strains that affect the development of CVDs, which will contribute to target the strains for treatment of CVDs in translational researches.
For example, the homologs with sequence coverage of less than 80% were filtered out [13]. Then, the obtained results were further processed with HMMER-Extractor (http://82.156.31.238:8000/index) [14]. The criteria for inclusion and exclusion were as follows: the intact metabolic gene cluster (MGC) of γBB should contain the six genes (caiTABCDE), and these genes must be from the same bacterial genome. In this case, the obtained strain-level genes should be the members of the intact caiTABCDE gene cluster. This pipeline has been proved to effectively reduce ambiguous annotations [14]. In addition, to avoid errors or omissions, we used the gutSMASH software [15] to directly annotate the MGCs of γBB in the genomes from NCBI RefSeq and Human Gut databases as mentioned above. After that, all the results were checked and merged into a core database. In the results generated by gutSMASH and HMMER-Extractor, the MGCs with the same genomic origin in the results are considered to be redundant, and only one result will be retained since each genome generally contains one γBB gene cluster.
species-level genome bins (SGBs) using metaWRAP, using the multiple k-mer sizes (parament: --k-list 21, 29, 39, 59, 79, 99, 119, 141) strategy [22]. Then, the microbial taxonomy was conducted based on the Genome Taxonomy Database (GTDB). Open reading frames (ORFs) were identified through the utilization of Prodigal v2.6.3 [23] and subsequently annotated by searching against the γBMGC using DIAMOND. The DIAMOND search was performed with an E-value threshold of ≤ 10-5 (option: -p 20). According to the previous researches, the comparison results were screened for ≥ 50% identity and 80% coverage (amino acids), which was a strict threshold for functional gene annotations [14, 24-25]. The abundance was calculated for each gene by using BBMap (https://jgi.doe.gov/data-and-tools/software-tools/bbtools/). The species abun-dance analysis was conducted through MetaPhlAn 4 [26].”
- Line 19: The phrase “comprehensive and accurate” could be expanded to explain how these qualities specifically benefit the analysis of γBB metabolism compared to other tools. Consider rephrasing to: “comprehensive and accurate, offering a higher resolution in identifying strain-specific γBB metabolic pathways compared to existing databases like KEGG or COG.”
Response: Thanks for your suggestion. Modifications made. For the details, see the text, please.
- Line 22: The accuracy metrics mentioned here are impressive, but the context for these metrics is missing. Add a brief comparison, such as: “achieving an annotation accuracy of 99.4%, which significantly outperforms existing databases like KEGG at similar thresholds.”
Response: Thanks. Modifications made. For the details, see the text, please.
- Line 39: The phrase “gut microbial dependent manner” would benefit from clarification. Suggest replacing it with: “in a manner dependent on gut microbiota activity, particularly involving species like Escherichia fergusonii and Proteus penneri.”
Response: Thanks. Modifications made. For the details, see the text, please.
- Line 49: The sentence “The conversion of γBB to TMA/TMAO intensified…” introduces an important concept but lacks supporting data. Suggest adding a reference to the relevant figure or supplementary table that confirms this observation.
Response: Many thanks for your suggestion, and we added a reference. The revised text reads as follows on lines 53-55:
“Furthermore, in the presence of γBB, the conversion of γBB to TMA/TMAO intensified, which implied the complexity of the "meta-metabolome" [3].
- R.A. Koeth, B.S. Levison, M.K. Culley, et al. Gamma-Butyrobetaine is a proatherogenic intermediate in gut microbial metabo-lism of L-carnitine to TMAO. Cell Metab. 2014; 20:799-812. doi: 10.1016/j.cmet.2014.10.006.”
- Line 69: The mention of “bacterial synthetic gene cluster caiTABCDE” would benefit from a diagram or figure reference explaining the cluster’s role in γBB metabolism. Consider suggesting: “Figure 1 illustrates the organization of the caiTABCDE gene cluster and its metabolic role in γBB synthesis.”
Response: Response: Thanks for your suggestion. Modifications made. The revised text reads as follows on lines 73-75:
“The organization of the caiTABCDE gene cluster and its metabolic role in γBB synthesis were illustrated in figure 1 as described below.”
Figure 1. The technical flowchart for the γBMGC database construction. The flowchart included data collection, sequence search (a), database validation and applications in metagenome (b). The homologs were searched in ~ 168,484 genomes with HMMER and gutSMASH. Then, the sequences were extracted through the HMMER-Extractor and Gut_extractor scripts, which were implemented with Python. Finally, all genes were checked and curated for constructing the γBMGC database. The best identity and accuracy test of γBMGC were conducted in a mock community and simulated gene dataset, respectively.
- Line 104: The reference to datasets accessed on August 11, 2023, should include additional information about dataset size and composition. Suggest including a parenthetical: (“comprising ~116,500 bacterial genomes from the NCBI RefSeq database”).
Response: Thanks for pointing this out. Modifications made.
- Line 126: The final database was curated using CD-HIT with 100% sequence identity. Explain why this threshold was selected, as stricter thresholds might exclude closely related sequences. Add a justification: “This threshold was chosen to eliminate redundancy while retaining unique strain-specific genes.”
Response: Many thanks for your suggestion and we have added the justification accordingly. The revised text reads as follows on lines 142-148:
“In the results generated by gutSMASH and HMMER-Extractor, the MGCs with the same genomic origin in the results are considered to be redundant, and only one result will be retained since each genome generally contains one γBB gene cluster. Representative amino acid sequences and nontarget homologs were deduplicated and clustered with 100% sequence identity through CD-HIT [16]. This threshold was chosen to eliminate redundancy while retaining unique strain-specific genes. Finally, all representative sequences and homologs were combined to construct the γBMGC database.”
- Line 138: In the validation formula section, explicitly describe the practical implications of metrics like PPV or NPV in annotating metagenomic datasets. For instance: “A high NPV ensures minimal false negatives, crucial for detecting rare but functionally significant γBB-related genes.”
Response: Many thanks for the reviewer's comments, and we have corrected the sentences accordingly.
- Line 186: The mention of 171,510 allelic genes lacks a direct comparison to other databases. Add a comparative statement: “This is significantly more than the 56 orthologs annotated in KEGG for the same metabolic pathway.”
Response: Many thanks and we have added the comparative statement accordingly.
- Line 230: When presenting the 50% identity cutoff results, explain why increasing identity thresholds leads to diminishing returns. Suggest adding: “Higher identity thresholds (e.g., 70%) reduced false positives but excluded sequences with natural variability, as seen in Figure 3c.”
Response: Many thanks. Modifications made.
- Line 256: The radar chart in Figure 4b is referenced but not described in sufficient detail. Suggest adding: “The chart highlights a sharp drop in detection rates at identity thresholds above 70%, underscoring the importance of balancing sensitivity and specificity.”
Response: Many thanks for the reviewer's comments, and we have added the sentence accordingly.
- Line 298: The statement that gene abundance is “significant (p < 0.05)” lacks statistical detail. Add the specific test used, such as: “as determined by the Wilcoxon rank-sum test, adjusted for multiple comparisons.”
Response: Many thanks. Modifications made.
- Line 309: The term “interesting” to describe Citrobacter spp.’s presence in multiple diseases is imprecise. Replace with: “notable, as Citrobacter spp. was consistently enriched across ACVD, HF, and hypertension samples.”
Response: Thanks. Modifications made.
- Line 318: The discussion of caiTABCDE’s role in CVD should reference a specific part of the results. Suggest: “As shown in Table S4, species with intact caiTABCDE clusters were predominantly associated with increased γBB levels.”
Response: Many thanks for your comments, and we have corrected the sentences accordingly.
- Line 376: The phrase “providing evidence at the level of specific genes” would benefit from clarification. Add a brief example: “such as the enrichment of Escherichia coli genes in ACVD samples (Figure 5b).”
Response: Response: Thanks for pointing this out. We have added a brief example: “such as the enrichment of genes from E. coli in ACVD samples (Figure 5b).”
- Line 397: The conclusion mentions “high accuracy” but does not reiterate key metrics. Consider adding: “with metrics such as 99.4% accuracy and 97.97% PPV validating its robustness.”
Response: Many thanks. Modifications made.
- Line 408: The GitHub link for data availability is mentioned, but the contents should be briefly described. Suggest: “This repository includes curated gene datasets, validation scripts, and the γBMGC database files.”
Response: Thanks. Modifications made.
- Line 428: Add: “All datasets were obtained from publicly accessible repositories with prior ethical approvals, as listed in Table S2.”
Response: Thanks. Modifications made.
- Figure 1 (Line 195): The flowchart lacks sufficient annotations. Add labels or notes explaining the key steps in the database construction pipeline, such as sequence extraction and HMM training.
Response: Thanks. We have redrawn figure 1 and added labels or notes explaining the key steps in the database construction pipeline.
Figure 1. The technical flowchart for the γBMGC database construction. The flowchart included data collection, sequence search (a), database validation and applications in metagenome (b). The homologs were searched in ~ 168,484 genomes with HMMER and gutSMASH. Then, the sequences were extracted through the HMMER-Extractor and Gut_extractor scripts, which were implemented with Python. Finally, all genes were checked and curated for constructing the γBMGC database. The best identity and accuracy test of γBMGC were conducted in a mock community and simulated gene dataset, respectively.

Reviewer 3 Report
Comments and Suggestions for Authors
Please see the attached.

Author Response
We thank the positive comments from the reviewer. Revisions are made to the manuscript in red color. We have redrawn figures with high resolution and descriptive legends in the revised manuscript, which can help readers understand how the database can be practically utilized. Thanks for your time and effort in helping us improve the manuscript and look forward to your final decision on publication.
This study showed the development of a comprehensive functional gene database which was designed for assisting analysis of cardiovascular diseases and understanding of microbial 1-carnitine pathway in human gut. Through the construction of dataset and validation of γBMGC database, it shows advantages to other orthology databases under the condition of more than 90% sequence identity as well as shorter run time. However, a much clearer explanations are needed especially for the data presented in multiple figures. In addition, the quality of all figures are poor which need to fix and more details are needed for figure descriptions.
Here are some comments:
- Line 25, please include the full name of CVD.
Response: Many thanks. We have added the full name of CVD (line 26: cardiovascular disease).
- Line 190-191, for those gene families whose sequences were not indexed by SwissProt or NCBI, you performed manual searches in the genomes of certain strains based on 50 literatures. Would you include more details about the manual searches? How to avoid human error when doing the searches? What is the cut-off to decide the gene families whose sequences were real for the study? Please explain.
Response: Many thanks for the reviewer's comments, and we have revised these sentences as follows on lines 219-222:
“The bacterial strains were reported to have the ability to synthetize γ-butylbetaine, and their genomes were downloaded. Then we made annotations for the MGCs with gutSMASH, which was conducted based on Hidden Markov Models with high accuracy. Afterwards, the candidate caiTABCDE gene clusters were scrutinized against the annotations to ensure the reliability of the developed database.”
- Line 197, figure 1 is quite complicated. However, the description for figure 1 is quite simple without telling details from sequence search, database validation to the finalize database. For example, what is the pfarm database in figure 1a? For figure 1b (line 205), what is the python script (perform a specific task) to filler out the other genes.
Response: Thanks for pointing this out. We have redrawn figure 1 and added details for the database construction. The Pfam database was a basis for gutSMASH, which contained reliable Hidden Markov Models. The python scripts were the HMMER-Extractor and Gut_extractor for sequence extraction, which could be checked on the GitHub website: https://github.com/xielisos567/caiTABCDE.
- Based on the description in line 201-203, there are 168,484 genomes for homologous genes, just want to know how many of them were manually checking? If the number is huge, how to make sure the manually checking process did not contain human error? Any datasets were used as references to confirm the accuracy? And why the human gut genomes were extended during the manually checking step instead of together with NCBI Refseq and human intestine. Or what is the purpose to extend the search against human Gut? Please explain. In addition, how the accuracy testing was performed? Please include more details to explain figure 1.
Response: Many thanks for your comments. We are sorry for the misunderstandings caused by our vague description. After sequence alignments in the genomes, we obtained a large number of candidate homologous genes. Then we made a statistic of these genes through the SeqKit tool for the gene length. As each gene of the caiTABCDE gene cluster has a reference gene length, we then check and filter these homologous genes based on the sequence coverage of these candidate genes. For example, genes with sequence coverage less than 80% compared to the reference genes will be filtered. In this case, finally we filtered ~ 1,566 genes cumulatively, leaving 171,510 genes that were matched with or close to the standard gene length.
In the actual operation process, the only difference is the order of expansion in different bacterial genomes. During the task execution, we found that the majority of the γBB gene cluster comes from human gut microbiome. Therefore, later we added ~ 47,000 genomes derived from human gut for the gene expansion. In fact, that is just a difference in chronological order, which is consistent with the manner of extending together in the NCBI RefSeq and human intestine databases. The accuracy testing of the γBMGC database was performed through a simulated gene dataset as described below. We have redrawn figure 1 and added detailed descriptions in the legends.
The revised text reads as follows on lines 236-251:
“Based on the seed dataset, we made a thorough search against ~ 168,484 genomes for the homologous genes (Figure 1a). After manually checking and filtering the homologs, ~ 1,566 unqualified genes were filtered and we finally obtained 171,510 allelic genes with each of them containing a specific strain origin (Figure 1b) (Table S3). This is significantly more than the 56 orthologs annotated in KEGG for the same metabolic pathway. Furthermore, with a 100% sequence identity cutoff, there generated 10,734 non-redundant γBB associated metabolic genes (Figure 1b) (Table S3). All the genes included nucleotide sequences and amino acids, ensuring functional gene annotations or abundance analysis at strain level. Notably, each of the obtained bacterial genome contained at least one caiTABCDE gene cluster composed of six metabolic genes in col-linearity, indicating that they worked together for synthesis of γ-butylbetaine. Finally, the constructed database was validated and applied in metagenome analysis (Figure 1b). When conducting metagenomic analysis, based on the γBMGC database, we could obtain the gene abundance (metabolic pathway) differences between healthy controls and the disease groups. Meanwhile, with MetaPhlAn 4, the disease-associated species could be further revealed, which could be combined with the gene abundance analysis.”
Figure 1. The technical flowchart for the γBMGC database construction. The flowchart included data collection, sequence search (a), database validation and applications in metagenome (b). The homologs were searched in ~ 168,484 genomes with HMMER and gutSMASH. Then, the sequences were extracted through the HMMER-Extractor and Gut_extractor scripts, which were implemented with Python. Finally, all genes were checked and curated for constructing the γBMGC database. The best identity and accuracy test of γBMGC were conducted in a mock community and simulated gene dataset, respectively.
- Line 222, figure 2, the x axis is too blurry to see the species. Except the first two species with high genus level, those species with low number of genomes at strain level would suggest to reset the scale (or zoom in the scale), otherwise, it would be too difficult to read the genus level. Please fix the figure.
Response: Many thanks. Modifications made.
Figure 2. Statics of strains at species (a) and genus level (b) in the γBMGC database. At species level, statistical analysis was conducted based on the top 20 bacterial species.
- Line 236-240, as the statement “increasing the identity from 20%-99% did not result in any reduction in sensitivity”. If this is the case, what would be the reason to use 50% identify and 80% coverage not 30% identify and 80% coverage? What is the advantages of using 50% identity comparing to others? If the total human samples increase, how it will impact the use of the % identify? Please clarify the statement in this paragraph.
Response: Thanks for your comments. We have clarified the statement in this paragraph. The added sentence was as follows on lines 282-285:
“Thus, although the sensitivity showed no fluctuations, the 50% identity and 80% coverage criteria exhibited a higher accuracy, PPV, specificity and F1-score than any other criteria such as the 30.0% identity cutoff, which was a suitable criterion for detection of the caiTABCDE genes.”
If the total human samples increase, we think that the usage of the % identify will not be impacted. The overall trend will be consistent with the analysis in this study because the benchmark in this study reflected a change trend in the functional gene’s detection, which was comparatively consistent with the existing researches.
- Line 246, Validation of γBMGC with a Mock Community: for the validation, it consisted 10 microbial genomes for the validation. It seems that 70% identity cut-off point was suitable for genome annotation. However, the accuracy, F1-score, and NPV were less than 98%. Would it related to the small microbial genomes that were used for validation? Line 266, figure 4a is too blurry to read the information. Please fix it.
Response: Thanks for your comments. We have redrawn figure 4 to make it clearer than the previous version.
For the validation, we believe that a certain deviation in accuracy, F1-score, and NPV is normal, which is related to the size, complexity, and assembly level of the genome to be analyzed. However, it does not affect the overall trend. The trend in this study is that the 70% threshold is a relatively reasonable and strict threshold for analyzing the γBB genes in microbial genomes.
- Line 278 to 282, this is the example under the condition of more than 90% sequence identity that the γBMGC database showed its advantage compared to other three database. However, when looking into the figure 4c, in the condition of 60%-80% sequence identity, the γBMGC database has shown higher false negative than COG and KEGG database, would this be a concern? Please explain the pros and cons.
Response: Many thanks for the reviewer's comments. The high detection rate of the γBMGC database under high threshold conditions indicates that its specificity is far superior to other available databases. However, the constructed γBMGC database could still have some deficiencies in species diversity, but these differences are tiny, which will not be a problem during data analysis. This is inevitable because our data source is the reference genome, and we do not adopt the assembled draft genomes from metagenomes such as the species-level genome bins (SGBs). These sketch genomes may come from completely new species and be included in other databases, but they do not meet the criteria for reference genes and therefore cannot be used for data analysis. In the future, we will further update the γBMGC database to ensure more comprehensive analysis. Thanks again for the valuable suggestions from the reviewer.
- Line 304, in figure 5, please also define the p-value of *, and ** in figure 5a. Is the HC (green one) the healthy control? Please include more details about the figure description. In addition, please use the high quality figure in the entire article. Figure 5b is also difficult to see the species name under each bar.
Response: Thanks for pointing this out. We have redrawn all the figures with high quality and provided more details about the figure 5 description. We also defined the p-value of *, and ** in figure 5a. The HC (green one) means the healthy control and we have added the details in the legend of figure 5. For the details, please see the text.
- Line 364-365, the statement “therefore, we recommend using a 50% identity cutoff combined with 80% coverage to analyze the caiTABCDE gene cluster in the metagenomic sequencing data.” This is confusing point which needs further explanation. Is the 50% identity cutoff only suitable for specifically case? Because in line 248-249, it said the result of this validation revealed all genomes were overestimated when using an identity cutoff 50%. Will this lower cut-off can lead to an increase in false positive results?
What is the expected result that was looking for here? Please explain.
Response: Many thanks for the reviewer's comments. We apologize for the misunderstanding caused. The description in lines 248-249 has been revised. We here aim to show that the 70% cutoff is a turning point, which means that below the 70% cutoff or even at 50% cutoff, all the genomes can be detected, and no false positives have been detected here. In addition, with the 50% identity (80% coverage) cutoff, the accuracy, PPV, specificity and NPV was 99.4%, 97.97%, 99.16% and 100%, respectively. In contrast, using a combination of 70% sequence identity and 80% coverage, all performance metrics, including accuracy, PPV, specificity and NPV, exceeded 96%. Obviously, the 50% identity (80% coverage) cutoff is a more suitable threshold for data analysis than the 70% identity cutoff. However, as we all know, a higher identity threshold usually means the lower false positive rate, in this case, the 70% identity cutoff could reduce false positives during data analysis, which in turn could be regarded as a relatively strict threshold. This is the conclusion obtained after comparing various performances comprehensively. At least in analysis based on the γBMGC database, the 50% identity cutoff was suitable for detecting most of the caiTABCDE genes in different samples.
- Line 397, for the conclusion, this is a database designed for assisting analysis of cardiovascular diseases and understanding of 1-carnitine pathway in human gut. Are there any limitations of this study?
Response: Many thanks for the reviewer's comments. We acknowledge that this study has certain limitations. As is well known, cardiovascular disease (CVD) has been proven to be directly related to components and metabolites from gut bacteria, such as trimethylamine-N-oxide (TMAO), phenylacetyl glutamine (PAGln), p-cresol, lipopolysaccharides (LPS) and N, N, N-trimethyl-5-aminovaleric acid (TMAVA), etc. In the future, more and more potential CVD associated bacterial metabolites will be discovered and reported. In this study, γ-butyrobetaine is a microbial intermediate metabolite that is the initial step of the l-carnitine → TMA pathway, which is the precursor of TMA (CVD related metabolite). It can be used as an indirect bacterial metabolite to establish associations with CVDs. Thus, establishing a γBMGC database can assist in analyzing the occurrence and development process of CVDs, providing basic data for targeted therapy of CVDs.

Round 2
Reviewer 2 Report
Comments and Suggestions for Authors/